# Nurse-assisted and multidisciplinary outpatient follow-up among patients with decompensated liver cirrhosis: A systematic review

**Malene Barfod O'Connell**[1]⊙*, **Flemming Bendtsen**[1]⊙, **Vibeke Nørholm**[2]‡, **Anne Brødsgaard**[3,4]‡, **Nina Kimer**[1]⊙

1 Gastrounit, Medical Division, Copenhagen University Hospital Amager Hvidovre, Hvidovre, Denmark,
2 Clinical Research Department, Copenhagen University Hospital Amager Hvidovre, Hvidovre, Denmark,
3 Department of Pediatrics and Adolescent Medicine, Copenhagen University Hospital Amager Hvidovre, Hvidovre, Denmark, 4 Department of Public Health, Section for Nursing, Aarhus University, Aarhus, Denmark

⊙ These authors contributed equally to this work.
‡ VN and AB also contributed equally to this work.
* malene.barfod.oconnell@regionh.dk

**Data Availability Statement:** All relevant data are within the paper and its Supporting information files.

## Abstract

### Background and objective

Liver cirrhosis represents a considerable health burden and causes 1.2 million deaths annually. Patients with decompensated liver cirrhosis have a poor prognosis and severely reduced health-related quality of life. Nurse-led outpatient care has proven safe and feasible for several chronic diseases and engaging nurses in the outpatient care of patients with liver cirrhosis has been recommended. At the decompensated stage, the treatment and nursing care are directed at specific complications, educational support, and guidance concerning preventive measures and signs of decompensation. This review aimed to assess the effects of nurse-assisted follow-up after admission with decompensation in patients with liver cirrhosis from all causes.

### Method

A systematic search was conducted through February 2022. Studies were eligible for inclusion if i) they assessed adult patients diagnosed with liver cirrhosis that had been admitted with one or more complications to liver cirrhosis and ii) if nurse-assisted follow-up, including nurse-assisted multidisciplinary interventions, was described in the manuscript. Randomized clinical trials were prioritized, but controlled trials and prospective cohort studies with the intervention were also included. Primary outcomes were mortality and readmission, but secondary subjective outcomes were also assessed.

### Results and conclusion

We included eleven controlled studies and five prospective studies with a historical control group comprising 1224 participants. Overall, the studies were of moderate to low quality,

**Funding:** The authors received no specific funding for this work.

**Competing interests:** The authors have declared that no competing interests exist.

and heterogeneity across studies was substantial. In a descriptive summary, the 16 studies were divided into three main types of interventions: educational interventions, case management, and standardized hospital follow-up. We saw a significant improvement across all types of studies on several parameters, but currently, no data support a specific type of nurse-assisted, post-discharge intervention. Controlled trials with a predefined intervention evaluating clinically- and practice-relevant endpoints in a real-life, patient-oriented setting are highly warranted.

## Introduction

Nurse-led and nurse-assisted outpatient care programs have proven safe and feasible across several chronic diseases. They have shown equal or better outcomes regarding health-related quality of life, symptom burden, and disease-specific clinical outcomes compared to physician-led programs [1]. A recent meta-analysis of four studies found no advantage of self-management programs on the clinical impact in patients with liver cirrhosis [2]. Still, the effect of nurse-assisted post-discharge interventions, including nurse-assisted multidisciplinary interventions among patients with decompensated liver cirrhosis, has not been investigated thoroughly.

Worldwide, liver cirrhosis represents a considerable health burden and causes 1.2 million deaths annually. The distribution of aetiologies and frequency among regions differs. Globally, Hepatitis C infection (HCV) is the leading cause of liver disease-related death, followed by Hepatitis B infection (HBV). In the western world, alcohol-related liver disease (ALD) and non-alcoholic fatty liver disease (NAFLD) are the most frequent causes and are increasing in prevalence [3, 4]. Liver cirrhosis evolves from compensated cirrhosis without any apparent signs of liver disease and no significant health issues related to the liver disease to decompensated cirrhosis characterized by several complications [5, 6]. Patients with liver cirrhosis are often diagnosed at a late stage, where the patients are decompensated with complications such as infections, ascites, hepatic encephalopathy, and variceal bleeding, all of which indicate a poor prognosis and severely reduced health-related quality of life [7–9]. Studies have shown that 20–37% of patients with liver cirrhosis are readmitted fewer than 30 days after hospitalization for decompensation, with a higher 90-day mortality rate than those who are not readmitted [10–12].

It has been recommended to engage nurses in the outpatient care of patients with liver cirrhosis [5, 6, 13]. At the decompensated stage, the treatment and nursing care are directed at specific complications, educational support, and guidance concerning preventive measures and signs of decompensation [5, 6]. Lifestyle changes and self-care are essential to disease management. Still, low compliance with prevention and treatment recommendations and inadequate self-care among patients with liver cirrhosis often leads to worse outcomes [14–16].

This review aimed to assess the effects of nurse-assisted and multidisciplinary follow-up interventions after admission with decompensation in patients with liver cirrhosis from all causes.

## Methods

A study protocol was registered at PROSPERO, the international prospective register of systematic reviews, and approved on May 15, 2019 (CRD42019128249). This systematic review is

reported according to the Preferred Reporting Items for Systematic Reviews and Meta-Analyses [17].

## Search strategy

We searched the following databases Pubmed, Embase, Cinahl, Web of Science, and Cochrane Libraries. Clinicaltrials.gov and the International Clinical Trials Registry Platform were searched for ongoing trials. Conference proceedings, reference lists for the included studies, and bibliographies were searched manually. All studies were reviewed irrespective of publication status, and no data range was applied. The following search terms were applied in different combinations and adjusted according to the specific database: "Liver cirrhosis,""Cirrhosis," "Nurs*," "Outpatient," "Aftercare," "Rehabilitation," "Intervention," "Follow-up," "Outpatient Care," "Post-discharge," "Post-admission," "Post hospitalization." The central search strategy is available in the S1 Appendix.

## Eligibility criteria and study selection

Studies were eligible for inclusion if i) they assessed patients diagnosed with liver cirrhosis by liver biopsy or classical clinical signs in combination with biochemistry and ultrasound, CT scan, Liver elastography, or other relevant imaging techniques, and had been admitted with a one or more complications to liver cirrhosis, and ii) if a nurse-assisted follow-up intervention including nurse-assisted multidisciplinary intervention was tested in the trial. The primary outcomes stated in the PROSPERO record were mortality, readmissions, and quality of life. We further included the secondary outcomes of self-efficacy, disease knowledge, lifestyle, and cost-effectiveness. Randomized clinical trials were prioritized, but non-randomized controlled trials and prospective cohort studies were also included due to low numbers. Case studies, reviews, expert opinions, and retrospective and registry studies were excluded, as studies with no clinically relevant outcomes and studies assessing interventions without a nurse-assisted element. Excluded studies and the reason for exclusion are presented in supplementary S2 Appendix.

## Data collection process

Two reviewers independently screened the searched titles, abstracts, full texts, and extracted data. All authors resolved disagreements regarding inclusion or exclusion or discrepancies in data extraction by consensus. The following data were extracted: study characteristics (author, country, year of publication, study design, funding source), patient demographics (age, sex, number of patients in each study arm, Child-Pugh score [18], and Model for End-stage Liver Disease (MELD) [19] scores, follow-up, type of intervention(s) and outcomes as defined by the authors of the studies.

## Risk of bias

The risk of bias for randomized trials was assessed using the Cochrane Handbook for Systematic Reviews of Interventions [20, 21], assessing the following bias items: i) selection bias (randomization, allocation), ii) performance bias (blinding of participants, personnel), iii) detection bias (blinding of assessors and outcomes), iv) attrition bias (incomplete outcome data), and v) reporting bias (selection of the reported results). The risk of bias for prospective studies with no control group was assessed using the Methodological Index for Non-Randomized Studies (MINORS) [22].

## Results

### Literature search

Electronic searches were performed in May 2019 and updated in February 2022. We identified 2741 studies (Fig 1). Manual searches identified another five studies of relevance. After screening titles and abstracts and removing duplicates, 123 studies were assessed. Letters and studies irrelevant to the aim of our study were excluded, and 31 studies were further evaluated. Nine studies assessed interventions after hospital discharge with liver cirrhosis complications [23–31], and seven assessed interventions among stable outpatients [14, 32–37]. These 16 studies, comprising 1224 participants, were included in a descriptive summary.

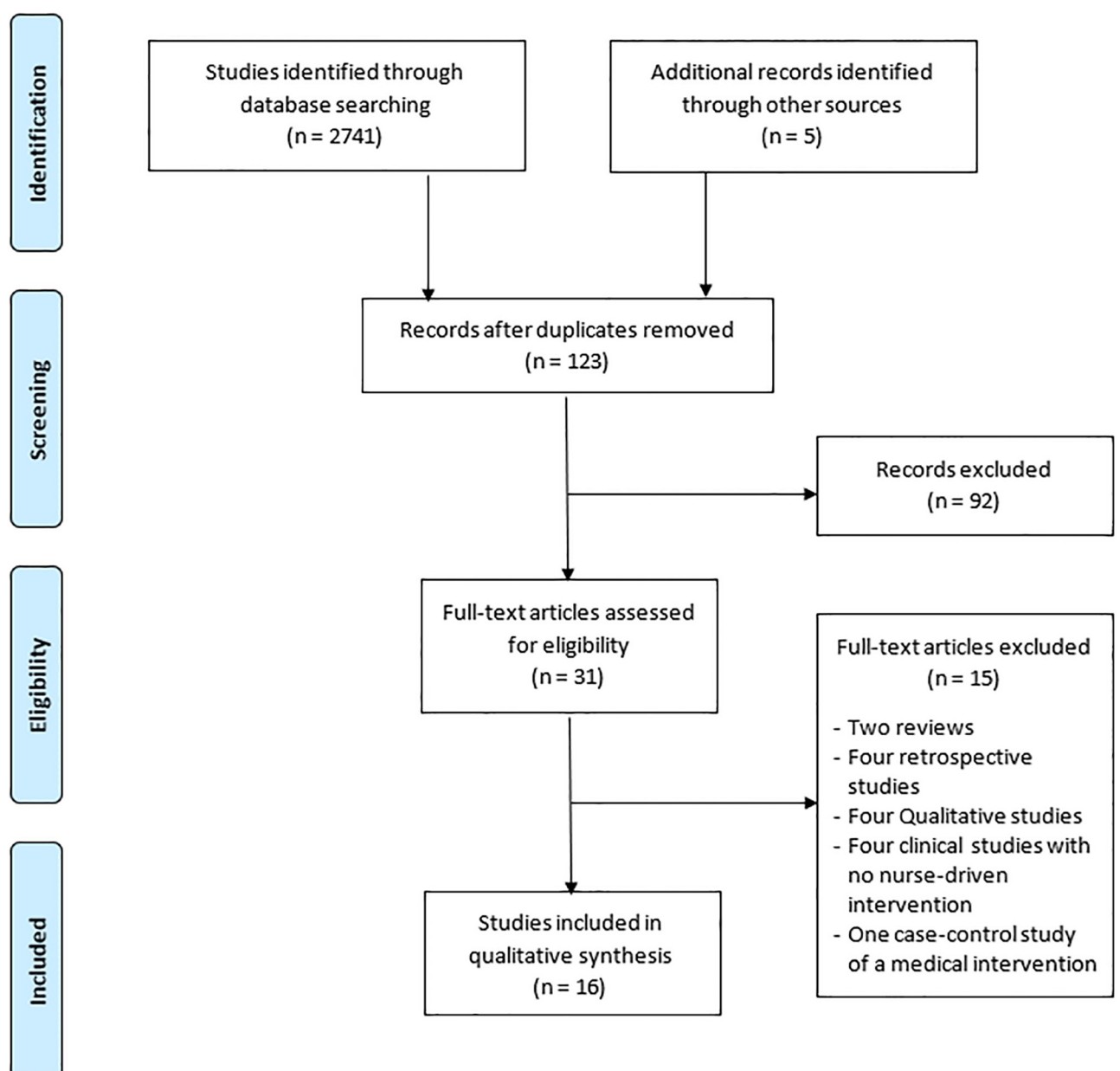

**Fig 1. Prisma flow diagram.**

| | Garrido | Mansouri | Morando | Sussman | Wigg | Zandi |
|---|---|---|---|---|---|---|
| Random sequence generation | + | + | ? | + | + | ? |
| Allocation concealment | - | - | - | - | + | - |
| Blinding of participants and personel | - | ? | - | - | - | - |
| Blinding of outcome assessment | - | + | - | ? | - | - |
| Incomplete outcome data | + | + | - | - | + | + |
| Selective reporting | + | + | ? | + | + | ? |
| Other bias | ? | + | ? | ? | + | ? |

**Fig 2. Bias assessment of the randomized controlled trials.** Guided by the Cochrane Handbook for Systematic Reviews of Interventions.

## Quality of evidence

A bias assessment of the randomized controlled trials is presented in Fig 2. In general, studies were open-label, and only one study performed blinding of outcome assessment [33]. One study conducted blinding of allocation [25]. One study had a 40% dropout rate after the intervention, and attrition bias was likely [23]. Non-randomized studies were evaluated by the Methodological Index for Non-Randomized Studies (MINORS) [22] (Fig 3). None of the comparative studies scored a maximum of 24 points, and none of the non-comparative studies scored a maximum of 16 points. All studies enrolled and collected data prospectively. None of the studies evaluated results blinded or carried out an unbiased assessment of study results. Only one trial reported prior sample size calculation [36]. Two studies had a loss to follow-up of 9% and 23% [14, 35]. Of note, one study was published as an abstract [30], and reporting bias was suspected. Based on the above, all studies were of low or moderate quality.

## Study characteristics

Study characteristics of the 16 included studies are presented in Table 1. The included studies were published between 2005 and 2019. Of the 16 studies included, six trials comprising 361

| | Andersen | Morales | Majc | Kumral | Alavinejad | Bajaj | Ganapathy | Goldsworthy | Volk |
|---|---|---|---|---|---|---|---|---|---|
| A clearly stated aim | 2 | 2 | 2 | 2 | 2 | 2 | 2 | 2 | 2 |
| Inclusion of consecutive patients | 2 | 2 | 2 | 2 | 2 | 2 | 2 | 2 | 2 |
| Prospective collection of data | 2 | 2 | 2 | 2 | 2 | 2 | 2 | 2 | 2 |
| Endpoints appropriate to the aim of the study | 2 | 2 | 2 | 2 | 2 | 2 | 2 | 2 | 2 |
| Unbiased assessment of the study endpoint | 0 | 0 | 0 | 0 | 0 | 0 | 0 | 0 | 0 |
| Follow-up period appropriate to the aim of the study | 0 | 2 | 2 | 1 | 2 | 2 | 1 | 2 | 2 |
| Loss to follow-up less than 5% | 2 | 2 | 2 | 2 | 1 | 2 | 2 | 2 | 1 |
| Prospective calculation of the study size | 0 | 0 | 0 | 0 | 0 | 2 | 0 | 0 | 0 |
| An adequate control group | 2 | 2 | 1 | 2 | | | | | |
| Contemporary groups | 1 | 1 | 1 | 1 | | | | | |
| Baseline equivalence of groups | 1 | 2 | 1 | 1 | | | | | |
| Adequate statistical analysis | 1 | 2 | 2 | 2 | | | | | |
| Total | 16 | 19 | 17 | 17 | 11 | 14 | 11 | 12 | 11 |

**Fig 3. Bias assessment of the non-randomized studies.** Guided by Methodological Index for Non-Randomized Studies.

**Table 1. Characteristics of the 16 included studies.**

| Study | Patient source | Patient description | Study design | No. of patients | Intervention | Control | Follow-up |
|---|---|---|---|---|---|---|---|
| **Randomized clinical trials** | | | | | | | |
| Garrido 2017, Italy [32] | Center for Liver Diseases, Padova University Hospital | Stable outpatients | RCT | 20 intervention 19 control | Education on HE, incl. basic information, bowel emptying, medicine, and monthly telephone calls | Standard care | 12 months |
| Mansouri 2016, Iran [33] | Transplantation Center Nemazee, Shiraz University of Medical Sciences | Stable outpatients on a waiting list for liver transplantation | RCT, researcher blinded | 40 intervention 40 control | Self-management training: sessions on diet, medicine, problem-solving, decision-making, cognitive-behavioral techniques, and empowerment | Standard care and a booklet | One month |
| Morando 2013, Italy [26] | General Hospital Padova | Admitted with acute complications to cirrhosis | Consecutive allocation after discharge to intervention or control by a 2:3 ratio | 40 intervention 60 control | Care Management Check-up. Including structured diagnostics, treatment, and follow-up after 1–12 weeks | Standard outpatient care by primary physician and 'on-demand' hepatologist | 9–11 months |
| Sussman 2005, USA [23] | Liver Disease Unit, Ranchos Los Amigos National Rehabilitation Centre, LA | Alcoholic liver cirrhosis admitted to Liver Disease Unit | RCT, pilot study | 13 intervention 12 control | Individual education: instruction on liver disease, nutrition, sanitation, motivation enhancement, and decision-making/personal commitment | Standard care, including referral to social service, a psychologist, or a dietician, if needed. | Three months |
| Wigg 2013, Australia [25] | Hepatology Unit of Flinders Medical Centre, Adelaide | Admitted with chronic liver failure-related complications | Randomized, controlled, parallel-group study design | 40 intervention 20 control | Home visit by nurse one week after discharge, weekly nurse telephone calls, and telephone reminders of appointments. Nurse visits involved decision-making and self-management support, diet and medication education | Standard inpatient, hospital outpatient, and primary care management | 12 months |
| Zandi 2005, Iran [34] | Tehran Hepatitis Center | Stable outpatients with controls during the inclusion period | Every other participant allocated to the treatment groups | 21 intervention 23 control | Educational sessions with relatives, including the nature of the disease, coping strategies, relaxation techniques, diet, nutrition, medicine, and two pamphlets. Telephone follow-up every 14 days | Standard care, no education given | Three months |
| **Study** | **Patient source** | **Patient description** | **Study design** | **No. of patients** | **Type of intervention** | **Type of control** | **Follow-up** |
| **Clinical trials with a historical control group** | | | | | | | |
| Andersen 2013, Denmark [24] | Abdominal center, University Hospital Bispebjerg | Alcoholic liver cirrhosis and admitted with HE | A prospective study with historic control from the same center | 19 intervention 14 control | A session with a nurse after discharge, alcohol school for some patients, and help with social services were offered | Standard outpatient control, as defined by a responsible physician | 20 months |
| Kumral 2015, USA [30] | The University of Virginia (hospital name not reported) | Liver cirrhosis and acute decompensating event | Prospective study, historic control from the same center | 20 intervention 25 control | Nurse teaching session with the patient and family members, booklet, a digital scale, and pill organizer, 72-hour post-discharge phone call | Standard care is not defined | One month |
| Majc 2018, Slovenia [27] | Department of Gastroenterology, Murska Sobota General Hospital | Liver cirrhosis and active drinkers of alcohol | Prospective study, historic control from other departments of the same hospital | 98 intervention 101 control | Education on alcohol abstinence, diet, and adjusting diuretic therapy. | No regular outpatient control | Five years |

*(Continued)*

**Table 1.** (Continued)

| Zhang 2019, China [31] | Department of gastroenterology and Yijishan Hospital, Wannan Medical college | Liver Cirrhosis | Prospective study, historic control from the same department | 30 intervention 30 control | Four-stage health education guided by empowerment theory. | One-to-one training and health education, including dynamic evaluations until patients were informed | Two months |
|---|---|---|---|---|---|---|---|
| Morales 2017, Spain [28] | Hepatology Unit, Germans Trias y Pujol Hospital, Badalona | Liver cirrhosis and discharged after admission with complications | Prospective study, historical control group from the same center | 80 intervention 112 control | HEPACONTROL program. 7-day follow-up visit after discharge with a hepatologist, medicine adjustment, lab tests, or diagnostics as needed; leaflet on warning signs of decompensation. | Outpatient visit within two months after discharge, no lab tests or diagnostics available | Seven months minimum |
| **Study** | **Patient source** | **Patient description** | **Study design** | **No. of patients** | **Intervention** | **Control** | **Follow-up** |
| **Prospective clinical trials** | | | | | | | |
| Alavinejad 2019, Iran [35] | Ahvaz Jundishapur University of Medical Sciences, Ahvas | Liver cirrhosis and referred to Research Center | Pre-post study design | 72 intervention | 2-hour session led by a physician, nurse, and nutritionist with face-to-face education on liver cirrhosis, nutrition, and lifestyle; educational booklet; weekly telephone calls from a nurse | None | Six months |
| Bajaj 2017, USA [36] | Division of Gastroenterology, Hepatology and Nutrition, Richmond | Outpatient liver cirrhosis and a co-dwelling caregiver | The prospective design of testing intervention | 20 patient/caregiver dyadic intervention | Weekly group therapy sessions and skills acquisition in A) Stress Management, B) Dealing with Depression, C) Adjusting to Anxiety, D) Family Health and Changes in Roles. | None | Five weeks |
| Ganapathy 2017, USA [29] | Division of Gastroenterology, Hepatology and Nutrition, Richmond | Liver cirrhosis admitted with complications and ready for discharge | Prospective proof of concept trial | 40 patient/caregiver intervention | Instruction in using the Buddy app for messaging and reporting symptoms; educational session on complications of cirrhosis and methods of action; brochure with telephone numbers and instructions; three visits and two telephone calls after discharge | None | 30 days |
| Goldsworthy 2017, UK [37] | Department of Hepatology, Leeds Teaching Hospitals | Liver cirrhosis, attending a hepatology outpatient clinic | Pre-post design of prospective cohort with intervention | 52 intervention | 12-minute video on the nature of liver cirrhosis | None | 1–6 months |
| Volk 2013, UK [14] | Division of Gastroenterology, University of Michigan Health System | Outpatients with liver cirrhosis at a liver transplant referral center | Prospective quality improvement study | 150 intervention | Booklet on prevention and management of complications and self-monitoring of medications, appointments, and weight | None | Three months |

HE: Hepatic encephalopathy; SUPPH: Strategies Used by People to Promote Health Questionnaire

participants were open-label randomized controlled trials [23, 25, 32, 33] or consecutive allocated trials [26, 34]. Another five trials, with 529 participants, used a historical control group [24, 27, 28, 30, 31]. Five studies comprising 334 participants were prospective cohort studies with pre- and post-intervention but no control groups, and these were included in the

descriptive summary of the outcomes [14, 29, 35–37]. Two studies targeted patients with alcoholic liver cirrhosis [23, 24], and four focused on complications such as hepatic encephalopathy and ascites [24, 26, 29, 30]. The follow-up period varied from 30 days to five years. In the 16 studies assessed, the number of participants ranged from 25 to 199. The mean age went from 40.0 to 65.3 years, with an age range of 20 to 79. Of the participants, 827 were male (67.6%); one study did not report age and sex [30]. Five studies were carried out in the USA, three in Iran, two in Italy, and one in Denmark, Spain, the United Kingdom, Slovenia, Australia, and China.

## Descriptive summary of interventions

The 16 studies were divided into three main types of interventions: i) educational interventions, where an educational tool (video, booklet, educational classes, mindfulness or empowerment) was used to benefit the patient [14, 23, 29–37]; ii) case management studies characterized by a collaborative approach used to assess, plan, facilitate and coordinate care to meet patient and family health needs and intended to improve individual outcomes [24, 25]; and iii) standardized hospital follow-up where a structured protocol was implemented [26–28]. For each of the 16 studies, the intervention is shortly described in Table 1.

**Educational studies.** Eleven studies assessed a variety of educational interventions and varied in both duration and design. Five studies used a single educational tool [14, 23, 32, 36, 37], while six used a multifaceted intervention with various tools [29–31, 33–35]. The educational tools included films, apps, booklets/pamphlets, individual education, and group classes. Three studies used mindfulness-, health empowerment- and self-efficacy theories as a basis for the patient education program [31, 33, 36]. Kumral et al. was the only study that divided participants into different educational groups depending on their academic level and time spent in the sessions [30]. Five studies included more than one session/visit [23, 29, 31, 33, 34]. In five studies, the patients were followed up or checked for adherence by telephone call [29–31, 34] or a combination of text messages and telephone calls [35]. Four studies involved caregivers chosen by the participant in the intervention [29–31, 36], but Bajaj et al. was the only study assessing caregiver-reported outcomes [36].

**Case management.** Two studies were case management studies [24, 25], with specialized nurses as the primary care providers, combined with assistance from multidisciplinary teams comprising physicians, dieticians, and alcohol consultants. Both interventions included individual patient and caregiver education, home visits, and referral for alcohol treatment if needed. Both studies used individualized patient action plans. Andersen et al. planned the patient interventions according to the etiology of cirrhosis and the patient's physical and social problems and needs [24]. Wigg et al. individualized the intervention to target the patient's specific complications, conducted weekly telephone follow-ups, and offered telephone consultations for concerned patients [25].

**Standardized hospital follow-up.** Three studies used a standardized hospital follow-up design [26–28]. In all three studies, the patients were followed by a multidisciplinary team consisting of nurses specializing in treating patients with liver diseases and hepatologists. Morando et al. designed a structured program consisting of an ultrasound of the liver, upper endoscopy, neurocognitive assessment, assessment of alcohol use, and updated medical and dietary treatments [26]. Morales et al. designed a program consisting of a seven-day follow-up, including a physical exam, one-on-one interviews to determine potential risk factors, adherence to treatment and diet recommendations, medicine adjustment, lab tests or diagnostics as needed, and a leaflet on warning signs of decompensation [28]. Majc et al. designed a structured follow-up with regular outpatient controls. The education was customized for each

patient and included information about alcohol cessation, diet, and adjustment of diuretic treatment. Every patient was offered alcohol treatment, and family members were asked to assist in recovery [27]. In all three studies, follow-up visits were planned according to the severity of cirrhosis [26–28].

## Intervention outcomes

Intervention endpoints, outcomes and measurements for the 16 individual studies are presented in Table 2.

In the following we will present the results from our primary endpoints, mortality, readmissions and quality of life, and our secondary outcomes of disease knowledge, self-efficacy, lifestyle, and cost-effectiveness.

**Primary outcomes.** *Mortality*. Five studies comprising two randomized studies [25, 26] and three clinical trials with a historical control group [24, 27, 28] presented mortality as an intervention outcome. The three standardized hospital follow-up studies all showed significantly reduced mortality rates [26–28]. Morando et al. showed lower all-cause mortality in the intervention group (23.1%) compared to the control group (45.7%) (p = 0.025) [26]. Majc et al. showed a median survival time of 4.66 years in the intervention group compared to 2.9 years in the control group (p = 0.021) [27]. Morales et al. showed significantly lower mortality at 60 days follow-up (p = 0.016) but no differences in 30 days (p = 0.134) or 90 days follow-up (p = 0.655) [28]. Between the two standardized hospital follow-up studies, Wigg et al. showed no difference in risk of death between the two groups (p = 0.32) [25], and Andersen et al. demonstrated a significantly higher survival rate in the intervention group (p = 0.012) [24].

*Readmissions*. Nine studies comprising three randomized studies [25, 26, 32], four clinical trials with a historical control group [24, 27, 28, 30], and two prospective clinical trials [29, 35] presented readmissions as an intervention outcome. The three standardized hospital follow-up studies showed diverse readmission results [26–28]. The randomized clinical trial by Morando et al. showed significantly lower 30-day and 12 months readmission rates and fewer days of hospitalizations in the intervention group (15.4%) compared to the control group (42.4%) (p<0.01) [26]. The clinical trial with a historical control group by Majc et al. showed a non-significant reduction in hospitalizations over five years with a lower average number of hospitalizations in the intervention group (1.88) compared to the control group (2.07) (p = 0.612) [27]. Another clinical trial with a historical control group by Morales et al. showed a significant reduction in early admissions (<30 days) in the intervention group compared to the control group (95% CI 19–75) (p = 0.003), longer time from discharge to early readmission in the intervention group (p<0.001), and a shorter duration of admissions in the intervention group (p = 0.007). There was no significant difference between the groups in readmissions after 30 days (p = 0.278) [28]. The educational studies showed mixed results [29, 30, 32]. The clinical trial with a historical control group by Kumral et al. showed a significantly reduced 30-day readmission rate in the intervention group (25%) compared to the control group (62%) (p = 0.02) [30]. The randomized clinical trial by Garrido et al. showed possible prevention of admissions for hepatic encephalopathy, but the study was not equipped to assess this outcome [32]. The prospective clinical trial by Ganapathy et al. measured both 30-day readmissions and HE-related readmissions. They showed that 17 patients (42.5%) were readmitted within 30 days, but none due to hepatic encephalopathy, and eight potential HE-related readmission were prevented through app-generated alerts [29]. Another prospective clinical trial by Alavinejad et al. showed a decrease in days of hospitalization (p = 0.001) [35]. The two case management studies by Wigg et al. and Andersen et al. showed no significant difference in hospital admissions [24, 25].

**Table 2. Intervention outcomes in the 16 included studies.**

| Study | Endpoints | Outcomes | Measurements |
|---|---|---|---|
| **Randomized clinical trials** | | | |
| Garrido 2017, Italy [32] | Increase in awareness of hepatic encephalopathy; prevention of hepatic encephalopathy | • May prevent admissions for HE, but the study was not equipped to assess this outcome<br>• Improvement in knowledge on HE management from 5% (95% CI 1 to 24) to 80% (95% CI 58 to 92) (p<0.001)<br>• 12 patients (60%) declared satisfaction with an educational tool (95% CI 39 to 78) | • Questionnaire on the Awareness of Encephalopathy (QAE)<br>• Psychometric hepatic encephalopathy score (PHES)<br>• Child-Pugh score<br>• Model of End-stage Liver Disease (MELD) |
| Mansouri 2016, Iran [33] | Self-efficacy | • Increase in total self-management scores in the intervention group compared to the control group (p<0.05) | • Strategies Used by People to Promote Health (SUPPH) |
| Morando 2013, Italy [26] | Mortality, readmissions within 30 days; Hospital readmissions; Mean hospital stay during follow-up; 12-month survival; Cost analysis | • Lower all-cause mortality in intervention group (23.1%) compared to control group (45.7%) (p = 0.025)<br>• Lower 30-day emergent readmission in intervention group (15.4%) compared to control group (42.4%) (p<0.01)<br>• Lower mean number of days of hospital stay per patient month life in the intervention group compared to the control group (p<0.025)<br>• Higher costs of specialized caregiver model for the intervention group than treatment in the control group.<br>• Lower overall management costs in the intervention group compared to the control group (p<0.05) | • Model of End-stage Liver Disease (MELD)<br>• Sequential Organ Failure Assessment (SOFA) |
| Sussman 2005, USA [23] | Effect on interest in treatment; knowledge; self-reported lifestyle changes | • Greater learning in the intervention group (P<0.004)<br>• Greater change in average lifestyle score in the intervention group (Time x Condition interaction effect F = 3.09, P<0.05) | • Knowledge of lifestyle behaviors by 19 forced-choice items<br>• Lifestyle assessed through 18 forced-choice items |
| Wigg 2013, Australia [25] | Liver-related occupied bed days; Admission rate; Length of stay; MELD and Child-Pugh scores; Quality of life. | • No difference in risk of death between groups (p = 95% CI, 0.3–1.5) (p = 0.32)<br>• No difference in occupied bed days between groups (p = 0.39)<br>• No difference in liver-related admissions between groups (p = 0.52)<br>• 30% higher attendance rate in the intervention group compared to the control group (p = 0.004)<br>• No change in the quality of life between groups at six months (p = 0.76) and 12 months (0.80)<br>• Significant improvements in the intervention group compared to the control group in HCC screening (p = 0.04), referral for liver transplant assessment (p = 0.048), the commencement of hepatitis A and B vaccination (p<0.001), bone density measure (p = 0.006) and vitamin D testing (p = 0.02)<br>• Increase of MELD score in the intervention group at six months (p = 0.02) and 12 months (p = 0.01)<br>• No difference in Child-Pugh scores between groups at six months (p = 0.49) and 12 months (p = 0.80) | • Chronic liver disease questionnaire (CLDQ)<br>• Quality of care indicators and questionnaire<br>• Satisfaction survey<br>• Child-Pugh score<br>• Model of End-stage Liver Disease (MELD) |
| Zandi 2005, Iran [34] | Quality of life; severity of disease | • No difference in the severity of liver disease between groups (p = 0.73)<br>• Increase in CLDQ in the intervention group (mean score 139 to 171.6) (p = 0.001)<br>• Decrease in CLDQ in the control group (mean score 137 to 112.2) (p = 0.001) | • Chronic liver disease questionnaire (CLDQ)<br>• Need assessment questionnaire<br>• Self-report questionnaire |

*(Continued)*

**Table 2.** (Continued)

| Study | Endpoints | Outcomes | Tools applied |
|---|---|---|---|
| **Clinical trials with a historical control group** | | | |
| Andersen 2013, Denmark [24] | Survival; Admissions; Alcohol consumption; Cost | • Improved survival in the intervention group compared to the control group (p = 0.012)<br>• No significant difference in readmissions between groups (p = 0.99)<br>• Decrease in alcohol use in the intervention group<br>• Similar cost for subsequent admissions in intervention and control group (0.65) | None stated |
| Kumral 2015, USA [30] | 30-day readmission rates | • Reduced 30-day readmission rates in the intervention group (25%) compared to the control group (62%) (p = 0.02).<br>• Relative risk reduction of 60% with a number needed to treat (NNT) of 2.7 to prevent one 30-day readmission | None stated |
| Majc 2018, Slovenia [27] | Hospitalizations; the number of outpatient examinations; survival; Alcohol consumption | • Higher survival probability in the intervention group (p = 0.021)<br>• Not significant lower number of hospitalizations in the intervention group compared to the control group (p = 0.612)<br>• Lower alcohol consumption in the intervention group compared to the control group (p = 0.018)<br>• Increased use of outpatient examinations in the intervention group (5.54 examinations) compared to the control group (2.27 examinations) (p = 0.000) | • Alcohol Use Disorders Identification test (AUDIT)<br>• CAGE questionnaires |
| Morales 2017, Spain [28] | Early readmission; Emergency department visits; Mortality at 30, 60, 90 and end of follow-up, Costs | • Lower mortality rate in the intervention group compared to the control group at 60 days follow-up (p = 0.016)<br>• No significant difference in mortality between groups at 30 days follow-up (p = 0.134), 90 days follow-up (p = 0.655), or at the end of follow-up (p = 0.510)<br>• No significant difference between groups in readmissions after 30 days (p = 0.278)<br>• Reduction in early readmission rate (<30 days) in the intervention group compared to the control group (95% CI 19–75) (p = 0.003)<br>• Longer time to the first readmission in the intervention group compared to the control group (p<0.001)<br>• Shorter duration of hospital stays in the intervention group compared to the control group (p = 0.007)<br>• Fewer emergency department visits due to cirrhosis-related complications in the intervention group compared to the control group (p = 0.035)<br>• Patients who had been readmitted early made more emergency department visits compared to those who were readmitted after 30 days (p = 0.002) | None stated |
| Zhang 2019, China [31] | The activity of daily living; health knowledge; Health behavioral level | • Higher HPLP II scores in the study group compared to the control group at discharge and two months after discharge (p<0.05)<br>• Greater total awareness rates in the intervention group compared to the control group (p = <0.05)<br>• Greater BI scores in the intervention group compared to the control group two months after discharge (p = 0.006) | • Health-promoting lifestyle profile II questionnaire (HPLP II)<br>• Health knowledge questionnaire<br>• Barthel index scores (BI) |

(*Continued*)

**Table 2.** (Continued)

| Study | Endpoints | Outcomes | Tools applied |
|---|---|---|---|
| **Prospective clinical trials** | | | |
| Alavinejad 2019, Iran [35] | Days of hospitalization; Quality of life; Knowledge about the disease; lab tests; clinical symptoms of decompensation | • Decrease in days of hospitalization (p = 0.001)<br>• Increase in disease knowledge scores (p<0.0001)<br>• Increase in CLDQ (p<0.0001)<br>• Decrease in no. of patients with edema (p = 0.002) and ascites (p = 0.005)<br>• No effect on MELD score (p = 0.552) | • Chronic liver disease questionnaire (CLDQ)<br>• Liver cirrhosis knowledge questionnaire |
| Bajaj 2017, USA [36] | Change in health-related quality of life; change in Beck Depression Inventory-II, Beck Anxiety Inventory, Sleepiness Scale, Sickness Impact Profile scores | • Improvement in overall SIP scores (p = 0.005)<br>• Improvement in BDI II scores (p = 0.012)<br>• No effect on BAI score (p = 0.51); PHES score (p = 0.75); MELD score (p = 0.48); ESS score (p = 0.13).<br>• Improvement in PSQI scores (p<0.001)<br>• Improvement in ZBI-SF scores (p = 0.04) | • Sickness impact profile (SIP)<br>• Beck Depression Inventory-II (BDI-II)<br>• Beck Anxiety Inventory (BAI)<br>• Pittsburg Sleep Quality Index (PSQI)<br>• Zarit Burden Interview Short Version (ZBI-SF)<br>• Perceived Caregiver-Burden (PCB) |
| Ganapathy 2017, USA [29] | 30-day readmissions; HE-related readmissions; Fall risk; the number of alerts from the app: medicines, sodium intake; weight; admissions; Evaluation of Buddy app | • 17 patients were readmitted within 30 days, none of these because of Hepatic encephalopathy (HE)<br>• Eight potential HE-related readmissions were prevented through app-generated alerts<br>• TUG times improved in patients without readmissions<br>• 1657 alerts in total were generated for missed critical medicines, missed sodium intake, and rapid increase in weight<br>• 114 alerts initiated by patients/caregivers requesting medical input | • Timed Up and Go test (TUG) |
| Goldsworthy 2017, UK [37] | Knowledge; Change in knowledge; Evaluation of educational video | • Low median baseline questionnaire score (25%) (IQR 16.7–41.7%)<br>• Improvement of 41.7% in median questionnaire scores at follow-up (p<0.001)<br>• Educational video was rated as useful and relevant | • Questionnaire covering complications and management of liver cirrhosis<br>• Screencast evaluation |
| Volk 2013, USA [14] | Patients' knowledge; knowledge improvement | • Low median score in baseline knowledge survey (53%)<br>• Improvement in median knowledge scores from 53% to 67% (p<0.001) | • A 15-item survey covering diet, over-the-counter medication, and health maintenance activities |

*Quality of life.* Four studies comprising two randomized clinical trials [25, 34] and two prospective clinical trials [35, 36] presented quality of life as an intervention outcome. Three studies measured health-related quality of life by the disease-specific questionnaire to assess the quality of life in patients with chronic liver disease Chronic Liver Disease Questionnaire (CLDQ). The 29-item questionnaire incorporates disease-specific and physical- and mental health questions into six domains [38]. The randomized clinical educational study by Zandi et al. showed an increase in CLDQ in the intervention group (mean score 139 to 171.6) (p = 0.001) and a decrease in CLDQ in the control group (mean score 137 to 112.2) (p = 0.001) [34]. The randomized clinical case-management study by Wigg et al. showed improved health-related quality of life scores in the intervention group during the study period but no significant improvement compared to the control group at six months (p = 0.76) and 12 months (0.80) [25]. The prospective clinical educational study by Alavinejad et al. showed an

increase in CLDQ in the intervention group (p<0.0001) [35]. Another prospective clinical, educational study by Bajaj et al. measured quality of life by the behaviourally based health status measure, The Sickness Impact Profile (SIP), which was designed to measure the extent to which health and illness affect daily life and functioning [36]. The results showed an improvement in the overall SIP score in the intervention group (p = 0.005) [36].

**Secondary outcomes.** *Disease knowledge*. Six educational studies comprising two randomized clinical trials [23, 32], one clinical trial with historical control group [31], and three prospective clinical trials [14, 35, 37] presented disease knowledge as an intervention outcome. The randomized clinical trial by Sussman et al. measured knowledge in the intervention group using 19 forced-choice items concerning liver cirrhosis, including preventive measures and complications. The study showed significantly greater learning in the intervention group P<0.004 [23]. Another randomized clinical trial by Garrido et al. measured knowledge in the intervention group using the Questionnaire on the Awareness of Encephalopathy (QAE) [32]. The study showed a highly significantly increased knowledge in the intervention group concerning the management of hepatic encephalopathy from 5% (95% CI 1 to 24) to 80% (95% CI 58 to 92) p<0.001 [32]. The clinical trial with a historical control group by Zhang et al. measured knowledge by a health knowledge questionnaire for liver cirrhosis developed for the specific study with answer options "know" or "don't know" [31]. The study showed significantly greater total awareness rates in the intervention group compared to the control group (p = <0.05) [31]. In the prospective clinical trial by Alavinejad et al., knowledge was measured by a liver cirrhosis knowledge questionnaire developed for the specific study with 20 questions with the answer options "know" or "don't know." The study showed that the knowledge scores improved significantly before (141.89 ± 20.40) and after (182.72 ± 10.27) the educational intervention (P < 0.0001) [35]. In another prospective clinical trial by Goldsworthy et al., knowledge was measured by a nine-item questionnaire concerning liver cirrhosis developed for this specific study. The study showed a baseline median questionnaire score of 25.0% (IQR 16.7–41.7%) which increased to 66.7% (IQR 50.0–75.0%) at follow-up, showing a significant improvement of 41.7% (p<0.001) [37]. The prospective clinical trial by Volk et al. measured knowledge using a 15-item survey covering topics such as diet, medications, and health maintenance activities. The study showed a significant median knowledge score improvement from 53% to 67% (p<0.001) [14].

*Self-efficacy and Health-promoting lifestyle*. One randomized educational study by Mansouri et al. measured self-reported self-efficacy by the 29-item Strategies Used by People to Promote Health (SUPPH) [33]. The study showed a significant increase in total self-efficacy scores after self-management training in the intervention group compared to the control group (p<0.05) [33]. One educational clinical trial with a historical control group by Zhang et al. measured health-promoting lifestyle by the health-promoting lifestyle profile II questionnaire (HPLP II) [31]. The study showed a significantly higher health-promoting lifestyle score in the intervention group compared with the control group at both discharge and two months after discharge [31].

*Lifestyle*. Four studies comprising two randomized clinical trials [23, 24] and two clinical trials with historical control groups [27, 31] presented lifestyle measures as an intervention outcome. The randomized controlled educational study by Sussman et al. measured lifestyle change using an 18 forced-choice questionnaire on diet, alcohol, smoking, drugs, exercise, and medication compliance. The study showed a more significant difference in average lifestyle score in the intervention group (P<0.05) [23]. The clinical educational trial with a historical control group by Zhang et al. measured health-promoting lifestyle using the 52-item Health-promoting lifestyle profile II questionnaire (HPLP II), including questions related to the six dimensions of health responsibility, exercise, nutrition, self-realization, interpersonal

relationships, and stress management [31]. The study showed significantly higher HPLP II scores in the study group compared to the control group at discharge and two months after discharge (p<0.05) [31]. Two clinical trials with a historical control group presented results on alcohol consumption [24, 27]. The case management study by Andersen et al. showed reduced alcohol consumption in the intervention group in 17 out of 19 participants, and five participants stopped alcohol consumption altogether. Data for the control group was unreliable and not used for statistical analysis [24]. The standardized follow-up study by Majc et al. showed a significant decrease in alcohol consumption in the intervention group compared to the control group (p = 0.018) [27].

*Cost-effectiveness.* Two studies conducted a cost-effectiveness analysis [26, 28]. The randomized controlled, standardized hospital follow-up study by Morando et al. showed higher costs of the specialized caregiver model for the intervention group than the treatment in the control group. However, the costs of readmissions were higher in the control group causing significantly lower overall management costs in the intervention group compared to the control group (p<0.05) [26]. The case-management study by Andersen et al. showed that the median economic costs for subsequent hospital admissions were similar in the intervention and control group (p = 0.65) [24].

## Discussion

Through a systematic approach to the search strategy and data quality assessment, this review summarises the best evidence available on mortality, readmissions, quality of life, self-efficacy, disease knowledge, lifestyle, and cost-effectiveness in cases of decompensated cirrhosis, where a nurse-assisted follow-up, including nurse-assisted multidisciplinary interventions, was administered. The results show three main types of interventions: educational interventions, case management studies, and standardized hospital follow-up. We saw significant improvement across all types of studies on several parameters, including objective outcomes such as mortality, readmissions and cost-effectiveness and subjective outcomes such as quality of life, self-efficacy, and patient knowledge. The three standardized hospital follow-up studies all showed significantly reduced mortality rates. We did though see mixed results concerning readmissions rates and quality of life across the different types of interventions, indicating that currently, no data support a specific type of nurse-assisted or multidisciplinary post-discharge intervention.

The present review shows that controlled trials with a predefined intervention evaluating clinically- and practice-relevant endpoints in a real-life, patient-oriented setting are highly warranted. As described in previous studies, there is a need for randomized clinical studies concerning standardized nursing care and chronic care models with multidisciplinary involvement in the treatment of liver cirrhosis [5, 39, 40], as the efficacy of such programs is widely shown in other major chronic diseases, including heart failure and chronic obstructive pulmonary disease [41, 42]. Future studies should focus on evaluating and validating nurse- and physician-driven clinics and programs of rehabilitation to secure personalized healthcare services for patients and their families and optimize the use of healthcare resources. The involvement of a patient's family in health care has proven beneficial in other chronic diseases [43, 44]. In a recent review concerning nursing care of patients with liver cirrhosis, the involvement of caregivers or family members is stated as an essential part of both treatment and follow-up [5]. Several studies show the positive effect and importance of utilizing family nursing and involving families in health care for acute and chronically ill patients [44, 45].

In a multi-center randomized trial, patients with liver cirrhosis are offered standard care or participation in a nurse-led clinic in addition in addition to standard care [46]. The primary

outcome will be the physical and mental health-related quality of life, while the secondary outcomes include readmissions and disease progress. Another mixed methods hybrid type I effectiveness-implementation study aims to demonstrate the effectiveness and implementation feasibility of using an order-set in routine patient care within eight hospital sites in Alberta [47]. And, in a large randomized multi-cite study a 28-day home-based multidisciplinary intensive liver optimization programme "LivR" aimed at improving 28-day mortality and reducing 30-day readmission will be tested compared to standard care [48]. These studies may provide substantial evidence about nurse-assisted personalized management of cirrhosis.

## Limitations

All included randomized studies were classified as having a high risk of bias, and all non-randomized studies were assessed as having low certainty of the evidence. Given the nature of the interventions in question, blinding participants and personnel was neither feasible nor rational, partly explaining the high risk of performance bias in the randomized studies. Some studies may have moderated performance bias by objective outcomes such as hospital admission and mortality. Detection bias could be a potential source of bias in the individual studies in this review due to many subjective outcomes, such as self-efficacy, lifestyle improvements, and quality of life. A high attrition rate is reported in several studies in this review [14, 23, 25, 28, 36, 37], which is expected in studies with vulnerable and frail populations. A substantial loss to follow-up could lead to both overestimations of the treatment effects and underestimation due to attrition of the most fragile patients [49]. Studies were conducted in countries that vary in culture, religion, and etiology of cirrhosis; thus, comparing results from different healthcare settings should be considered. Lastly, concerning the multifaceted educational interventions, it should be considered which part of the educational program created the beneficial effects or if it was the combination of the educational program and closer contact with health care personnel that proved effective. Future studies with stratified outcomes may improve the evidence of using educational tools in post-discharge follow-up.

Initially, we planned to perform meta-analyses of mortality outcomes and the number of patients admitted to the hospital during the follow-up period. Due to the heterogeneity of the eight randomized studies concerning populations, interventions, controls, and outcomes, a meta-analysis was not found justified to perform [50].

## Conclusion

In this review, we applied a systematic approach to nurse-assisted and multidisciplinary outpatient follow-up among patients with decompensated liver cirrhosis. Studies were found to be of low to moderate quality. We saw significant improvement across all types of studies on several parameters, including objective outcomes such as mortality, readmissions, and cost-effectiveness, and subjective outcomes such as quality of life, self-management, and patient knowledge, but currently, no data support a specific type of nurse-assisted, post-discharge intervention. Due to the low number of randomized studies included in this systematic review, divergent methodology, and high heterogeneity across interventions, the present review shows that controlled trials with a predefined intervention evaluating clinically- and practice-relevant endpoints in a real-life, patient-oriented setting are highly warranted.

## Supporting information

**S1 Appendix. The central search strategy.**
(PDF)

**S2 Appendix. Excluded studies and the reason for exclusion.**
(PDF)

**S1 Checklist. PRISMA 2020 checklist.**
(DOCX)

## Acknowledgments

The authors wish to thank their colleagues, Steve Sussman, Alan Wigg, Bojan Tepes, and Dejan Majc, for answering questions about their research and, where possible, providing additional data on outcomes. Special thanks to Frank V. Schiødt and Sarah Montagnese for providing individual patient data from their trials conducted in Denmark and Italy, respectively.

## Author Contributions

**Conceptualization:** Malene Barfod O'Connell, Flemming Bendtsen, Nina Kimer.

**Formal analysis:** Malene Barfod O'Connell, Nina Kimer.

**Investigation:** Malene Barfod O'Connell.

**Methodology:** Malene Barfod O'Connell, Nina Kimer.

**Project administration:** Malene Barfod O'Connell, Nina Kimer.

**Resources:** Malene Barfod O'Connell.

**Supervision:** Flemming Bendtsen, Vibeke Nørholm, Anne Brødsgaard, Nina Kimer.

**Writing – original draft:** Malene Barfod O'Connell, Nina Kimer.

**Writing – review & editing:** Malene Barfod O'Connell, Flemming Bendtsen, Vibeke Nørholm, Anne Brødsgaard, Nina Kimer.

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
