## [Decision Letter · Decision Letter 0]

2 Oct 2022

PONE-D-22-22848Nurse-assisted and multidisciplinary outpatient follow-up among patients with decompensated liver cirrhosis: A systematic reviewPLOS ONE

Dear Dr. OConnel,

Thank you for submitting your manuscript to PLOS ONE. After careful consideration, we feel that it has merit but does not fully meet PLOS ONE’s publication criteria as it currently stands. Therefore, we invite you to submit a revised version of the manuscript that addresses the points raised during the review process.

Dear Authors,

the topic is extremely interesting. Despite this, the reviewers have raised several issues (especially methodological) that I fully agree with.

Major revisions of the manuscript are therefore required in order to be reconsidered for publication on Plose one.

Please submit your revised manuscript. Please include the following items when submitting your revised manuscript:A rebuttal letter that responds to each point raised by the academic editor and reviewer(s). You should upload this letter as a separate file labeled 'Response to Reviewers'.A marked-up copy of your manuscript that highlights changes made to the original version. You should upload this as a separate file labeled 'Revised Manuscript with Track Changes'.An unmarked version of your revised paper without tracked changes. You should upload this as a separate file labeled 'Manuscript'.

We look forward to receiving your revised manuscript.

Kind regards,

Riccardo Nevola, MD, PhD

Academic Editor

PLOS ONE

Journal Requirements:

2. Please upload a new copy of Figures 2 and 3 as the detail is not clear. Please follow the link for more information: https://blogs.plos.org/plos/2019/06/looking-good-tips-for-creating-your-plos-figures-graphics/" https://blogs.plos.org/plos/2019/06/looking-good-tips-for-creating-your-plos-figures-graphics/

Reviewers' comments:

Reviewer's Responses to Questions

**Comments to the Author**

1. Is the manuscript technically sound, and do the data support the conclusions?

Reviewer #1: Yes

Reviewer #2: Yes

Reviewer #3: Partly

Reviewer #4: Yes

2. Has the statistical analysis been performed appropriately and rigorously? 

Reviewer #1: Yes

Reviewer #2: N/A

Reviewer #3: No

Reviewer #4: N/A

3. Have the authors made all data underlying the findings in their manuscript fully available?

Reviewer #1: Yes

Reviewer #2: No

Reviewer #3: Yes

Reviewer #4: Yes

4. Is the manuscript presented in an intelligible fashion and written in standard English?

Reviewer #1: Yes

Reviewer #2: Yes

Reviewer #3: Yes

Reviewer #4: Yes

5. Review Comments to the Author

Reviewer #1: Patients with decompensated liver cirrhosis develop a number of complications that result in high morbidity and mortality and frequent readmission, require constant and rigorous monitor both in and outside the hospital. Nursing care to both hospitalized and nonhospitalized patients is most important to help manage and prevent complications of the disease and improve quality of life by providing medical education to patients and caregivers. This comprehensive systemic review provide the valuable clinical practice evidence about the effect of nurses assisted outpatient follow up and multidisciplinary interventions. So I am pleased to recommed the editor to accept this paper.

Reviewer #2: Here, authors determined the effects of nurse-assisted follow-up after admission with decompensation in patients with liver cirrhosis from all causes. Overall, the study was performed well, but there are several concerns that should be addressed by the authors, as following:

- Major concerns

o In the literatures, there are several systematic reviews that analyzed the nurse-assisted and multidisciplinary out-patients follow-up to patients with liver cirrhosis. This systematic review is missing of novelty.

o Authors did not perform any statistical analysis in the following study.

- In introduction, authors should explain better that there are two stage of liver cirrhosis (compensated and decompensated) as well as two different approaches to care these patients

- Search strategy: authors should explain more clearly (i) which data range they choose, (for instance: all the databases that they use from 1992 to 2022) and (ii) why they choose that range.

- Result paragraph: data are presented in very descriptive way. They should be more focused on the comparison between groups (controlled trials vs allocated trials) and describe in more detail way what they discovered.

- Figure legend is missing

Reviewer #3: Dr. O'Connell and others made a systematic review on "Nurse-assisted and multidisciplinary outpatient follow-up among patients with decompensated liver cirrhosis" by searching the databases Pubmed, Embase, Cinahl, Web of Science, and Cochrane Libraries, 75 Conference proceedings, reference lists, and bibliographies (manually). There are included eleven controlled studies and five cohort studies comprising 1224 participants. The conclusions suggested positive effects of nurse-assisted and multidisciplinary follow-up for outpatients with 44 liver-cirrhosis. The meta-analysis method applied, the data, and the results look like very interesting. On the contrary, the results were reached based on the multi parameters of different categories, the criteria to classify the data are very confused, and very difficult to be justified to get the conclusions. There are no quantified data to show the significant differnece for the comparisons among the randomized, non-randomized patients and other groups with and without the nurse-assisted and multidisciplinary outpatient follow-up. Overall, this meta data analysis must be adjusted to be more accurate, precise, and the paper be more academic readable, no ambiguity. It is much appreciated if the authors could classify the parameters with some grades/levels and do further statistical analysis and show the solid results. The final solid conclusions are expected for your further analysis if possible.

Reviewer #4: Malene Barfod O’Connell and co-authors have done a systematic review on the highly relevant topic of nurse-assisted and multidisciplinary follow-up among patients with liver cirrhosis. The manuscript is well written and presents the systematic review in a comprehensible way. The systematic review is based on a protocol published prior to the conduct of the review and a PRISMA 2020 checklist is included in the submission. The authors identified sixteen studies comprising 1224 participants through a comprehensive literature search strategy. The quality of the included studies was low to moderate with mixed results with regards to the interventions impact on outcomes which in turn led to the conclusion that no data currently supports a specific type of nurse-assisted post-discharge intervention. Furthermore, a pre-planned meta-analysis was not justified due to the heterogeneity of the included randomized studies. In the final conclusion of the manuscript the authors nevertheless state that the review indicated that the nurse-assisted intervention could positively affect the selected objective outcomes. The devil’s advocate might conclude the opposite conclusion.

In an effort to assess the overall quality of the systemic review I have identified a few points listed below. Scoring the systematic review according to the AMSTAR-2 tool (Shea BJ et al. BMJ. 2017 Sep 21;358:j4008) yields a low quality score.

1. The PROSPERO protocol

a. The PROSPERO record is not matching the methods described in the manuscript (e.g. language restrictions on included manuscripts, the health care personnel considered (nurses vs. physicians) and intervention types).

b. Deviations from the protocol should be explained in the manuscript.

c. The PROSPERO record apparently (as of 28-SEP-2022) has not been updated with regards to the stage of the review.

2. Search strategy:

a. It is not described whether the reference lists of the included studies were included in the search

b. It is not described whether trial registries were included in the manual search

c. I cannot find a list of the excluded studies with an explanation of why they were excluded

6. PLOS authors have the option to publish the peer review history of their article (what does this mean?). If published, this will include your full peer review and any attached files.

Reviewer #1: No

Reviewer #2: No

Reviewer #3: No

Reviewer #4: **Yes: **Peter Nissen Bjerring

---

## [Author Response · Author response to Decision Letter 0]

26 Oct 2022

Dear Academic Editor and Reviewers. 

Thank you for your comments and recommendations. 

We have responded to each of the points raised by the reviewers in the attached "Response to reviewers".

---

## [Decision Letter · Decision Letter 1]

18 Nov 2022

Nurse-assisted and multidisciplinary outpatient follow-up among patients with decompensated liver cirrhosis: A systematic review

PONE-D-22-22848R1

Dear Dr. Malene Barfod OConnell,

We’re pleased to inform you that your manuscript has been judged scientifically suitable for publication and will be formally accepted for publication once it meets all outstanding technical requirements.

Kind regards,

Riccardo Nevola, MD, PhD

Academic Editor

PLOS ONE

Reviewers' comments:

Reviewer's Responses to Questions

**Comments to the Author**

1. If the authors have adequately addressed your comments raised in a previous round of review and you feel that this manuscript is now acceptable for publication, you may indicate that here to bypass the “Comments to the Author” section, enter your conflict of interest statement in the “Confidential to Editor” section, and submit your "Accept" recommendation.

Reviewer #1: All comments have been addressed

Reviewer #2: All comments have been addressed

Reviewer #3: All comments have been addressed

Reviewer #4: All comments have been addressed

2. Is the manuscript technically sound, and do the data support the conclusions?

Reviewer #1: Partly

Reviewer #2: Yes

Reviewer #3: Partly

Reviewer #4: Yes

3. Has the statistical analysis been performed appropriately and rigorously? 

Reviewer #1: I Don't Know

Reviewer #2: N/A

Reviewer #3: No

Reviewer #4: Yes

4. Have the authors made all data underlying the findings in their manuscript fully available?

Reviewer #1: Yes

Reviewer #2: Yes

Reviewer #3: Yes

Reviewer #4: Yes

5. Is the manuscript presented in an intelligible fashion and written in standard English?

Reviewer #1: Yes

Reviewer #2: Yes

Reviewer #3: Yes

Reviewer #4: Yes

6. Review Comments to the Author

Reviewer #1: The author using a systematic approach to nurse-assisted and multidisciplinary outpatient follow up among patients with decompensated liver cirrhosis. Due to divergent methodology and high heterogeneity across interventions in this systematic review studies, showed mixed results concerning readmission rates and mortality across the different types of interventions . In this review, the authors see significant outcomes improvement after nurse assisted multidisciplinary interventions but need further randomized studies to validate this findings.

Reviewer #2: (No Response)

Reviewer #3: This menu reviews a lot of data upon the published papers and have no competivive innovation and the clinical significance, which are not so required in one reviewed paper. I agree the paper to be published although it is not completely done with the statistical analysis because of the data complication as mentioned in the point to point responses. It would be appreciated if the authors could describe this limitation in the menu.

Reviewer #4: All comments have been addressed properly in the revised manuscript. I have no further comments.

7. PLOS authors have the option to publish the peer review history of their article (what does this mean?). If published, this will include your full peer review and any attached files.

Reviewer #1: No

Reviewer #2: No

Reviewer #3: No

Reviewer #4: **Yes: **Peter Nissen Bjerring

---

## [Editor Report · Acceptance letter]

24 Nov 2022

PONE-D-22-22848R1 

Nurse-assisted and multidisciplinary outpatient follow-up among patients with decompensated liver cirrhosis: A systematic review 

Dear Dr. O’Connell¹:

I'm pleased to inform you that your manuscript has been deemed suitable for publication in PLOS ONE. Congratulations! Your manuscript is now with our production department. 

Kind regards, 

on behalf of

Dr. Riccardo Nevola 

Academic Editor

PLOS ONE